# Bacterial Cross-Transmission between Inanimate Surfaces and Patients in Intensive Care Units under Real-World Conditions: A Repeated Cross-Sectional Study

**DOI:** 10.3390/ijerph19159401

**Published:** 2022-07-31

**Authors:** Elisabetta Kuczewski, Laetitia Henaff, Anne Regard, Laurent Argaud, Anne-Claire Lukaszewicz, Thomas Rimmelé, Pierre Cassier, Isabelle Fredenucci, Sophie Loeffert-Frémiot, Nagham Khanafer, Philippe Vanhems

**Affiliations:** 1Unité d’Hygiène, Epidémiologie et Prévention, Hôpital Edouard Herriot, Groupement Hospitalier Centre, Hospices Civils de Lyon, 69003 Lyon, France; anne.regard@chu-lyon.fr (A.R.); nagham.khanafer@chu-lyon.fr (N.K.); philippe.vanhems@chu-lyon.fr (P.V.); 2Public Health, Epidemiology and Evolutionary Ecology of Infectious Diseases (PHE3ID), Centre International de Recherche en Infectiologie (CIRI), Inserm, U1111,Université Claude Bernard Lyon 1, CNRS, UMR5308, ENS de Lyon, 69007 Lyon, France; laetitia.henaff@chu-lyon.fr; 3Service de Médecine Intensive—Réanimation, Pavillon H, Hôpital Edouard Herriot, Hospices Civils de Lyon, 69003 Lyon, France; laurent.argaud@chu-lyon.fr; 4Service de Réanimation, Pavillon P, Hôpital Edouard Herriot, Hospices Civils de Lyon, 69002 Lyon, France; anne-claire.lukaszewicz@chu-lyon.fr (A.-C.L.); thomas.rimmele@chu-lyon.fr (T.R.); 5Plateau de Microbiologie Environnementale et Hygiène Hospitalière, Laboratoire de Biologie et Sécurité de l’Environnement, Institut des Agents Infectieux, Hôpital de la Croix-Rousse, Groupement Hospitalier Nord, Hospices Civils de Lyon, 69004 Lyon, France; pierre.cassier@chu-lyon.fr (P.C.); isabelle.fredenucci@chu-lyon.fr (I.F.); 6Laboratoires Anios—Ecolab Company, 59260 Lezennes, France; sophie.loeffert@ecolab.com

**Keywords:** environment, hospital, ICU, high-touch surfaces, bacteria, contamination, hospital-acquired infection

## Abstract

**Background/Objectives**: Contaminated surfaces play an important role in the nosocomial infection of patients in intensive care units (ICUs). This study, conducted in two ICUs at Edouard Herriot Hospital (Lyon, France), aimed to describe rooms’ microbial ecology and explore the potential link between environmental contamination and patients’ colonization and/or infection. **Methods:** Environmental samples were realized once monthly from January 2020 to December 2021 on surfaces close to the patient (bedrails, bedside table, and dedicated stethoscope) and healthcare workers’ high-touch surfaces, which were distant from the patient (computer, worktop/nurse cart, washbasin, and hydro-alcoholic solution/soap dispenser). Environmental bacteria were compared to the cultures of the patients hospitalized in the sampled room over a period of ± 10 days from the environmental sampling. **Results:** Overall, 137 samples were collected: 90.7% of the samples close to patients, and 87.9% of the distant ones were positives. Overall, 223 bacteria were isolated, mainly: *Enterococcus faecalis* (15.7%), *Pantoea agglomerans* (8.1%), *Enterobacter cloacae/asburiae* (6.3%), *Bacillus cereus* and other *Bacillus* spp (6.3%), *Enterococcus*
*faecium* (5.8%), *Stenotrophomonas maltophilia* (5.4%), and *Acinetobacter baumannii* (4.9%). Throughout the study, 142 patients were included, of which, *n* = 67 (47.2%) were infected or colonized by at least one bacterium. In fourteen cases, the same bacterial species were found both in environment and patient samples, with the suspicion of a cross-contamination between the patient–environment (*n* = 10) and environment–patient (*n* = 4). **Conclusions:** In this work, we found a high level of bacterial contamination on ICU rooms’ surfaces and described several cases of potential cross-contamination between environment and patients in real-world conditions.

## 1. Introduction

Transmission of healthcare-associated infections in intensive care units (ICUs) occurs frequently via the hands of healthcare workers (HCWs) and contaminated environments, including medical equipment, electronic devices, and high-touch surfaces [1].

In ICUs, the microbial contamination on inanimate surfaces close to patients was described, while data on areas further away from patients and their clinical relevance are less known [2]. We conducted a prospective repeated cross-sectional study to evaluate the microbial contamination of inanimate surfaces both close to and distant from patients, as well as equipment in two ICUs in a French University Hospital. The aim of this study was to describe the isolated pathogens from surfaces under real-world conditions, in particular, the following four bacteria: *Staphylococcus aureus*, *Enterococcus faecium*, *Acinetobacter baumannii*, and *Pseudomonas aeruginosa*. We decided to focus our attention on these four bacteria, since they have been widely documented to survive on environmental ICU surfaces [1,2,3]. Moreover, they have been included in European standards as model microorganisms (*S. aureus* and *P. aeruginosa* in the standard EN16615 and E. *faecium* in the standard EN13727 for temperatures >40 °C, due to its resistance). *A. baumanii* has been introduced in the more recent standard EN 17272, due to its increasing role in hospital surfaces contamination.

Moreover, we explored the potential link between environmental contamination and colonization and/or infection in critically ill patients, in order to provide deeper information on this major issue.

## 2. Materials and Methods

### 2.1. Study Design

This repeated cross-sectional study was conducted in one room of two different ICUs at the Edouard Herriot Hospital, a 1160-bedded university hospital in Lyon, France, from January 2020 to December 2021.

### 2.2. Microbial Sampling

Environmental samples from room surfaces have been carried out with sterile non-woven wipes moistened with buffered peptone water with 10% neutralizer, in accordance with Standard 18593 (Dominique Dutscher). Samplings were performed once monthly, for 14 months, in two different ICUs (ICU#1 and ICU#2). ICU#1 is a 10-beds unit, with an average of approximately 570 hospitalizations per year, of which, 74.1% were hospitalized for more than 48 h. ICU#2 is a 15-beds unit, with an average of 1000 admissions per year and length of stay of over than 2 days for 69% of them. ICU#1 is a polyvalent ICU with surgical and medical activities, and ICU#2 is a medical unit.

One room was randomly chosen in each ICU for environmental samplings and remained the same all along the study. Their surfaces were of 13.4 m^2^ and 20.6 m^2^, respectively. In each room, five environmental samples were realized, depending on ICU room configuration. Among them, we distinguished between surfaces with highly probable patients’ contact (bedrails in ICU#1 and ICU#2; bedside table in ICU#1 and room-dedicated stethoscope in ICU#2) and HCWs’ high-touch surfaces, distant from patients (computer, worktop/nurse cart, washbasin, and hydro-alcoholic solution/soap dispenser) (Figure 1). The delay between the last cleaning of the room and sampling was noted. Samples were collected between 15 and 45 h after the last room cleaning (22.2 ± 6.0 h, on average). Surface disinfection was performed with a product composed of chlorure de didécyldiméthylammonium and n-(3-aminopropyl)-n-dodécylpropane-1,3-diamine (according to the bio-cleaning protocol applied in the hospital (Quality Documentary Management Protocol, version 1, 26 November 2012). Samplings took place irrespective of current room occupancy. In 85.7% of the cases, the patient was present in the room at the time of sampling. Nobody else was in the room during sampling, except for 6 occasions, with the presence of 1 or 2 staff members. Patients’ date of admission/discharge and room allocation were documented; patient cultures were obtained based on routine care and compared against environmental cultures from the same room. The bacterial species found in environmental samples (and their antibiograms for the microorganisms of interest) were compared with those isolated from patients hospitalized in the same rooms over a period of ±10 days [3,4]. Overall, the medical records (in particular, microbiological test results) were checked for 72 patients in ICU#1 and 70 patients in ICU#2.

#### Microbiological Analysis

The wipes sent to Laboratory of Environmental Microbiology were incubated for 48 h at 36 °C in tryptone soy broth, which was placed into a stomacher (for 30 s, speed 2). Successively, the samples were plated on four culture media targeting the main pathogens; subculture agars were MacConkey (Thermo Fisher Scientific) for *A. baumannii*; Cetrimide (BioMérieux) for *P. aeruginosa*; ChromID (BioMérieux) for *S. aureus*; bile-esculin agar (BioMérieux) for *E. faecium*. Species identification was performed with standard microbial methods (MALDI-TOF Vitek MS BioMérieux). The sample was considered positive when at least one bacterium (of interest or not) was detected. Nosocomial pathogens of particular interest (*S. aureus*, *E. faecium*, *A. baumannii*, and *P. aeruginosa)* were conserved, and their antibiotic susceptibility was checked (Vitek II BioMérieux).

### 2.3. Data Collection

Environmental data were collected from the internal informatic portal Hybrid^®^ used at Hospices Civils de Lyon. Clinical data were collected from Easily^®^ and ICCA^®^ (IntelliSpace Critical Care and Anesthesia, version H.02.01, Philips Healthcare, Andover, MA, USA) softwares (license allocated to Hospices Civils de Lyon). All clinical and microbiological data were entered into a Microsoft Excel database (Microsoft Corp., Redmond, WA, USA). The collected variables included: (i) information regarding environmental samplings (isolated microorganisms and their susceptibility pattern for the targeted bacteria, date, hour, and location of samplings, delay from the last surfaces cleaning, presence (or not) of a patient at the time of sampling); and (ii) information regarding patients who occupied these rooms ±10 days within sampling (date and hour of entry and discharge, as well as microbiological status at entry and during hospital stay). In the evaluation of potential cross-transmissions between environment–patient and patient-environment, previous patient carriage and bacteria with different susceptibility patterns (when available) were excluded. Patients received specific information regarding this study.

In order to evaluate the diffusion tendency of microorganisms in a room, we introduced the spreading coefficient (SC), which we defined as the ratio of the number of bacteria per sampling site to the number of bacteria per room. No spreading in the room corresponds to a SC = 1, while a diffusion is indicated by a SC > 1 and proportional to SC value. In order to minimize the risk of a misinterpretation of the germ diffusion in the room, we performed some tests to exclude a possible carrying on sampler hands. We obteined the same results of diffusion on several sampling sites of the rooms for the bacterial species mentioned above, with these two different sampling protocols: (i) change of gloves and disinfection with hydro-alcoholic gel between each environemental sample; (ii) no gloves wearing and hands disinfection with hydro-alcoholic gel after each sample. Moreover, we performed a control test of gloves at the end of a series of samplings in a room realized with the same pair of gloves. Gloves were sampled with a wipe and sent to the laboratory.

Statistical analysis (Shapiro–Wilk normality test and Student *t*-test) was performed with STATA 17 Base Reference Manual, College Station, TX: Stata Press. StataCorp. 2019.

## 3. Results

During this repeated cross-sectional study, 137 samples were taken in total from the two ICU rooms, comprising 83 samples from areas distant from patients and 54 samples from areas close to patients. The percentages of positive samples were 89.7% in ICU#1 and 88.4% in ICU#2. Altogether, 90.7% of close samples and 87.9% of distant ones were positive; 55.7% of positive samples were poly-microbial, and 44.3% were mono-microbial; in total, 223 bacteria were isolated.

Among all the samples, thirty-five (25.5%) contained at least one of the target bacteria (*n* = 17 (25%) in ICU#1 and *n* = 18 (26.1%) in ICU#2): *E. faecium* (*n* = 13, 5.8% of all the bacteria), *A. baumannii* (*n* = 11, 4.9%), *S. aureus* (*n* = 6, 2.7%), and *P. aeruginosa* (*n* = 5, 2.2%). Their susceptiblity pattern is shown in Table 1; no multi-drug-resistant (MDR) bacteria were found.

The other main bacteria found in the environmental samples were (number and percentage of the total number of microorganisms): *Enterococcus faecalis* (*n* = 35, 15.7%); *Pantoea agglomerans* (*n* = 18, 8.1%); *Enterobacter cloacae/asburiae *(*n* = 14, 6.3%); *Bacillus cereus* and other *Bacillus* spp (*n* = 14, 6.3%); *Stenotrophomonas maltophilia* (*n* = 12, 5.4%); *Staphylococcus epidermidis* (*n* = 9, 4.0%); *Pseudomonas putida* (*n* = 8, 3.6%); *Klebsiella pneumoniae* (*n* = 6, 2.7%); *Pseudomonas oryzihabitans* (*n* = 6, 2.7%); *Staphylococcus haemolyticus* (*n* = 5, 2.2%); *Enterobacter hormaechei* (*n* = 5, 2.2%); *Leclercia adecarboxylata* (*n* = 5, 2.2%); unidentifiable gram-negative bacilli (*n* = 5, 2.2%); *Escherichia coli* (*n* = 4, 1.8%); *Escherichia vulneris* (*n* = 4, 1.8%); *Stenotrophomonas rhizophila* (*n* = 4, 1.8%); *Staphylococcus non aureus* (*n* = 3, 1.3%); *Hafnia alvei* (*n* = 3, 1.3%); *Serratia marcescens* (*n* = 3, 1.3%); and *Pseudomonas fluorescens* (*n* = 3, 1.3%).

The most contaminated sites were: bedrails (100% of positive samples, 57 different microorganisms), computer keyboard and mouse (92.9% of positive samples, 55 different microorganisms), and bedside table (92.9% of positive samples, 22 different microorganisms). The most represented bacterial species, with respect to the sampling sites, are shown in Figure 2.

The SC of collected bacteria was: (i) equal to one for 48.9% of the microorganisms (no spreading); (ii) within one and two for 22.2% of them (intermediate spreading); and (iii) greater than two for 28.9% of them (high level of spreading). The bacteria with SC ≥ 2 were: *H. alvei* and *S. marcescens* (3.0); *E. faecium* (2.6); *E. hormaechei* (2.5); *E. faecalis* and *E. cloacae/asburiae* (2.3); *A. baumannii* (2.2); *B. cereus* and other *Bacillus* spp.; and *S. maltophilia*, *K. pneumoniae*, and *E. coli* (2.0). The test carried out on gloves at the end of a series of samplings indicated no sampler hands carrying: the microorganisms found on the room surfaces were not found on the gloves (only the bacterium *S. rhizophila* was found).

The main bacteria found in patients’ positive cultures were: *E. coli* (*n* = 21, 13.9%), *S. aureus* (*n* = 13, 8.6%), *S. epi* (*n* = 11, 7.3%), *P. aeruginosa* (*n* = 10, 6.6%), *C. albicans* (*n* = 10, 6.6%), *E. cloacae* (*n* = 8, 5.3%), *K. pneumoniae* (*n* = 8, 5.3%), and *E. faecalis* (*n* = 8, 5.3%). The main sources were: blood cultures (32.0%), *cytobacteriological examination of urine (21.1%)*, endotracheal suctioning (16.4%), broncho alveolar lavage/mini broncho alveolar lavage (7.0%), bronchial suctioning (4.7%), expectorations (4.7%), rectal swab (3.9%), orthopaedic harvesting (3.1%), pus or abscess sampling (3.1%), and biological fluid (2.3%).

The investigation of a potential cross-transmission environment–patient and vice-versa showed that fourteen bacteria (6.3% of the total) were found both in the environmental and clinical samples of the patients hospitalized in the room within the time interval of ± 10 days. Among them, we suspected: (i) ten cases of potential contamination patient-environment, in which the same bacterium was found first in clinical and then in environmental samplings (average delay 3.9 ± 3.4 days, median 3 days) (Figure 3 and Table 2, from 1 to 10); (ii) four cases of potential transmission environment–patient, in which the same bacterium was found first in the environment and then in patient clinical samples (average delay 5.8 ± 1.9 days, median 6.5 days) (Figure 3 and Table 2, from 11 to 14). We did not find a significant difference between the delay of the contaminations from the environment to the patient versus the contaminations from the patient to the environment (*p* = 0.19). Similarly, the delay from the last cleaning of the room did not show any difference, regarding potential cross-contamination or not (21.0 ± 2.8 h, median: 22.0 h, *n* = 12, vs. 23.1 ± 7.6 h, median: 22.0 h, *n* = 16, p = 0.37).

Interestingly, we described a case of potential bacterial acquisition from a prior room occupant (Figure 3 and Table 2, episodes 10 and 11). Here, we provide a detailed description of this observation, since we found it interesting. On 18 September 2021, in ICU#1, *E. cloacae* was present in the endotracheal suctioning of patient P#1, who was hospitalized in the room from 17–20 September. The environmental sampling of this room, performed on 21 September, revealed the presence of the same bacterium on the computer keyboard, mouse, and bedside table. After one week, the same microorganism was found in the endotracheal suctioning of patient P#4, who was hospitalized in the same room from 27–29 September and not carrying it before. Therefore, an acquisition from environment can be suspected for this patient.

## 4. Discussion

In this work, we found a high level of bacterial contamination on inanimate surfaces and noncritical medical equipment in ICU rooms, in particular, on bedrails [4]. The predominant isolates found in the environment were Gram-positive bacteria *E. faecalis*, *B. cereus* and other *Bacillus* spp., and *E. faecium*, as well as Gram-negative bacteria *P. agglomerans*, *E. cloacae/asburiae*, *S. maltophilia*, and *A. baumannii*, in accordance with published data in the United States and Europe [1,2,3]. We showed several cases of potential bacterial cross-transmission between inanimate surfaces and patients in ICU rooms, thus confirming the idea that the surfaces near patient bed are a reservoir of microorganisms, which can be opportunistic, but also pathogenic and responsible of infection for critically ill patients [5].

These bacteria can survive for very long periods on the surfaces, depending on the type of material, as described by Kramer and colleagues in 2006 [6] and, more recently, in 2021 by Wissmann and colleagues [7]. These periods, which can reach several months on plastics for many of them, have been estimated in laboratory conditions. Nevertheless, even in real-world conditions, these pathogens, incorporated into biofilms, can survive for long periods and resist cleaning/disinfection [8].

Interestingly, no MDR bacteria were found, which supports the idea that, despite the long-lasting survival of the majority of pathogens on surfaces, the mechanisms of cross-resistance between biocides and antibiotics are not predominant [9].

Importantly, we found that similar bacteria were isolated from wipes taken from several sampling sites in the room. This spreading was observed in particular in some bacterial species: *Enterococci* (*faecalis* and *faecium*)*, E**nterobacters (cloacae/asburiae* and *hormaechei, H. alvei*, *S. marcescens*), and *A. baumannii*. Multi-site presence of the same microorganism of frequently touched surfaces suggests HCWs’ hands contamination [10], considering the predominance of immobilization condition of ICU patients. Moreover, surfaces not accessible to patients, in particular, the keyboard and mouse, showed a high level of contamination [11], supporting staff hands carrying. Additionally, among the reported cases, we were able to describe a potential contamination from prior room occupant [12,13]. In summary, we can distinguish these different sources of surfaces contamination: the patient himself in the room, other patients in the ward who contaminated HCWs’ hands (or gloves) or shared equipment’s, and HCWs’ themselves.

Interestingly, beyond the bacteria of interest, some other pathogens were recurrent on ICU surfaces, such as *E. faecalis,*
*P. agglomerans*, *E. cloacae/asburiae, B. cereus* and other *Bacillus* spp, *S. maltophilia*, and *S. epidermidis.* Since many of them are frequently found in clinical context, too, we are planning to add some of these bacteria in future studies. To strengthen these results, it would also be interesting to target other ICUs of our hospital.

The limitations of this study were that: (i) it comported only one sampling per month and targeted bacteria, but neither viruses nor fungi; (ii) it was interrupted three times, due to waves of COVID-19 pandemic in France (in March–June 2020, October–December 2020, and April–June 2021); (iii) we checked antibiotic susceptibility for only four bacterial species of interest found in the environment; (iv) clinical bacteria were isolated during patient routine care and not specifically for this work, and their genome sequencing was not systematically realized.

In spite of that, we were able to find out on high-touch surfaces near patient bed several bacteria that play an important role, from a clinical point of view. To further confirm this point, we report here some recent results of the surveillance of ICU healthcare-associated infections, conducted in our hospital for more than 20 years. The nosocomial infections targeted in this surveillance are pneumonia and bloodstream infections. Data for the year 2021 are being sent at the time of writing this article. In 2020, the incidence was 7% for pneumonia and 2% for bloodstream infections. The main microorganisms identified by the 2020 surveillance were: (i) for nosocomial pneumonia: *P. aeruginosa* (20.0%), *K. pneumoniae* (14.5%), *E. aerogenes* (9.1%), *E. coli* (7.3%), *E. cloacae* (5.5%), and *S. aureus* (5.5%); (ii) for hospital-acquired bloodstream infections: *K. pneumoniae*, *P. aeruginosa* and *S. epidermidis* (14.3% for all of them), *E. cloacae*, *E. faecalis*, and *E. faecium* (7.1% for all of them).

As described in this paper, these microorganisms were also found on ICU room surfaces, which, therefore, represent a source of bacterial acquisition for patients, in particular, for critically ill ones.

All these observations clearly indicate that standard precautions, such as hand hygiene and improved recommendations or national guidelines for the cleaning of high touch surfaces, represent the major measures to control the transmission of pathogens at the hospital. A proper use of personal protective equipment (PPE), which are highly used in ICU to reduce the risk of exposure to contaminated body fluids, has also to be seriously considered with this aim. An incorrect use of them increases the risk of contamination of the HCWs’ skin and clothing, which can contribute to pathogen transmission and surface contamination [14].

The routine sampling of environmental surfaces in healthcare environments is not usually indicated, while it might be interesting, in order to identify an environmental source of infection/contamination and demonstrate the efficacy of disinfection or cleaning procedures in endemic or epidemic situations [15]. Presently, there is no standard method for measuring the cleanliness of surfaces and achievement of certain cleaning parameters or defining the level of microbial contamination that correlates with good or poor environmental cleaning practices. The standardization and improvement of disinfection practices [16], as well as the full respect of hygiene measures, are necessary to minimize the risk of environmental contamination and healthcare-associated infections.

## 5. Conclusions

This study provides an original contribution to the theme of environmental contamination, since it focuses on the link between environmental and clinical samples in real-world conditions. This allowed us to assess the risk of infection or colonization for patients hospitalized in ICU, due to environmental contamination, as well as the adequacy of cleaning protocols. Additionally, through this work we could collect many bacterial strains present on ICU rooms’ surfaces. Our future perspective is to investigate their susceptibility to disinfectants and the emergence of a possible mechanism of cross-resistance and/or reduction in susceptibility.

## Figures and Tables

**Figure 1 ijerph-19-09401-f001:**
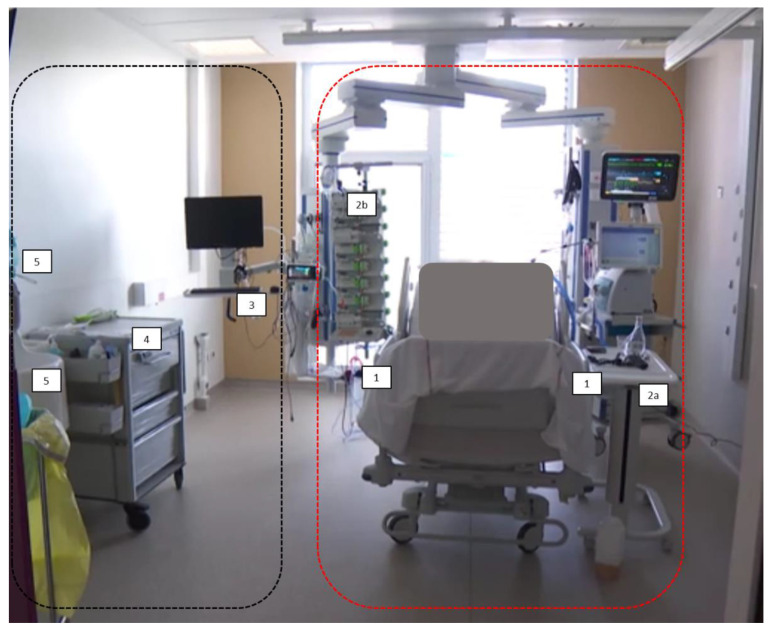
In an ICU room, sampling sites inside the red dashed line belong to the area close to patient (1, 2a in ICU#1, 2b in ICU#2), while sampling sites inside the black dashed line are in the zone distant from patient (3, 4, and 5). 1: foot-side bedrails; 2a: bedside table; 2b: room-dedicated stethoscope; 3: computer keyboard and mouse; 4: worktop/nurse cart; 5: washbasin and levers of hydro-alcoholic solution and soap dispensers.

**Figure 2 ijerph-19-09401-f002:**
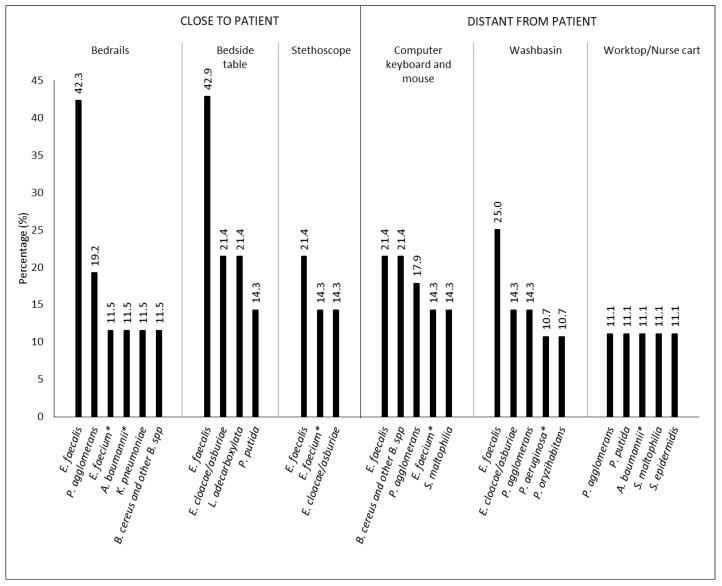
Distribution of bacteria found on room surfaces of ICU#1 and ICU#2, between January 2020 and December 2021 (*n* = 223), depending on sampling site. Only occurrences > 10% were shown (*: bacteria of interest).

**Figure 3 ijerph-19-09401-f003:**
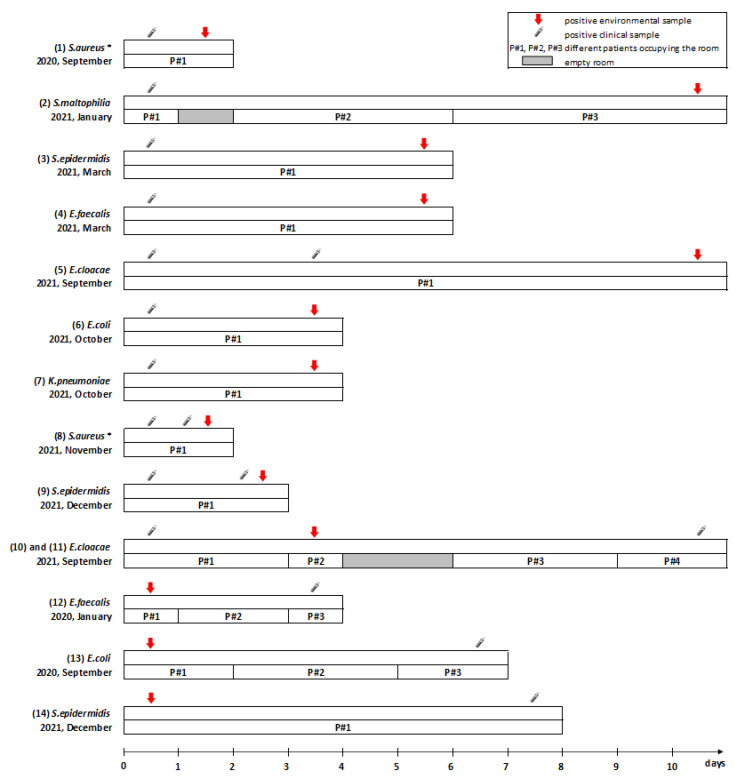
Synoptic representation of the cases of potential contamination (*n* = 14). Symbols are explained in the upper right legend (*: bacteria of interest); episodes 10 and 11 represent two potential cross-contaminations of the same bacterium from P#1 to environment and from environment to P#4.

**Table 1 ijerph-19-09401-t001:** Susceptibility pattern of the bacteria of interest (*n* = 35) (R: resistant, S: sensitive; I: intermediate; /: not tested).

** *A. baumannii* **	**Ticarcillin**	**Ticarcillin + clavulanic acid**	**Ceftazidime**	**Cefepime**	**Gentamicin**	**Ciprofloxacin**	**Levofloxacin**	**Cotrimoxazole**	**Aztreonam**	**Imipenem**	**Meropenem**
*n* = 11	S	S	S	S	S	S	S	S	R	S	S
** *P. aeruginosa* **	**Ticarcillin**	**Ticarcillin + clavulanic acid**	**Ceftazidime**	**Cefepime**	**Amikacin**	**Ciprofloxacin**	**Piperacillin**	**Piperacillin + Tazobactam**	**Aztreonam**	**Imipenem**	**Meropenem**
*n* = 1	S	S	S	S	S	R	S	S	S	R	I
*n* = 1	R	R	S	S	S	S	S	S	S	S	S
*n* = 3	S	S	S	S	S	S	S	S	S	S	S
** *S. aureus* **	**Oxacillin**	**Ofloxacin**	**Cotrimoxazole**	**Nitrofurantoin**	**Gentamicin**	**Vancomycin**	**Teicoplanin**	**Fusidic acid**	**Erythromicin**	**Tetracycline**	**Rifampicin**
*n* = 6	S	S	S	S	S	S	S	S	S	S	S
** *E. faecium* **	**Ampicillin**	**Levofloxacin**	**Cotrimoxazole**	**Nitrofurantoin**	**Gentamicin**	**Vancomycin**	**Teicoplanin**	**Linezolid**	**Quinupristin/dalfopristin**	**Tetracycline**	**Rifampicin**
*n* = 1	R	R	R	S	I	S	S	S	I	/	/
*n* = 1	R	R	I	S	I	S	R	S	I	/	/
*n* = 1	R	R	I	S	R	S	R	S	I	/	/
*n* = 1	R	R	I	S	I	S	S	S	S	/	/
*n* = 1	R	R	R	R	I	S	S	S	S	/	/
*n* = 2	R	R	R	S	I	S	S	S	S	/	/
*n* = 6	S	S	I	S	I	S	S	S	S	/	/

**Table 2 ijerph-19-09401-t002:** Potential contaminations from the patient to the environment (from 1 to 10) and from the environment to the patient (from 11 to 14). The delay was calculated with respect to the environmental sampling; therefore, it was negative when patient sampling preceded it and positive when patient sampling followed it (CBEU: *cytobacteriological examination of urine*; IUC: indwelling urinary catheter; AC: *arterial catheter*; CVC: *central venous catheter*; ETS: endotracheal suctioning; BAL: bronchoalveolar lavage; AC: *arterial catheter*) (bacteria of interest (*) were all susceptible).

Bacteria	Environmental Sample	Clinical Sample	Potential Contamination Delay (d: Days; h: Hours)	Cleaning Delay (Hours)	ICU#
(1) *S. aureus* *	Washbasin	Peripheral blood culture	−1 d	22.0	2
(2) *S. maltophilia*	Bedrails	Orthopaedic harvesting	−10 d	21.5	2
(3) *S. epidermidis*	Nurse cart	Peripheral blood culture	−5 d	22.0	2
(4) *E. faecalis*	Bedrails, bedside table, washbasin	CBEU (IUC)	−5 d	21.2	1
(5) *E. cloacae*	Bedrails, stethoscope	ETS; blood culture (CVC)	−10 d; −7 d	23.0	2
(6) *E. coli*	Bedrails, bedside table, washbasin	BAL	−3 d	22.2	1
(7) *K. pneumoniae*	Bedrails, washbasin	BAL	−3 d	22.2	1
(8) *S. aureus* *	Bedrails	Peripheral blood culture; expectorations	−1 d; −1 h	18.0	1
(9) *S. epidermidis*	Bedside table	Blood cultures (AC)	−2 d; −3 h	24.7	1
(10) *E. cloacae*	Keyboard/mouse, bedside table	ETS	−3 d	17.7	1
(11) *E. cloacae*	Keyboard/mouse, bedside table	ETS	7 d	17.7	1
(12) *E. faecalis*	Bedrails	CBEU (IUC)	3 d	22.0	2
(13) *E. coli*	Washbasin	CBEU (IUC)	6 d	14.7	1
(14) *S. epidermidis*	Nurse cart	Blood culture (AC)	7 d	23.0	2

## Data Availability

Not applicable.

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
