# Peer review of "Bacterial Cross-Transmission between Inanimate Surfaces and Patients in Intensive Care Units under Real-World Conditions: A Repeated Cross-Sectional Study"

_ijerph, 2022, doi:10.3390/ijerph19159401_

Round 1

Reviewer 1 Report

Thank you for this interesting addition to the body of work for environmental presence and patient crossover of relevant organisms.  I enjoyed reading about your work.

Consider adding the number of patients included (n=142) and how many had positive cultures (n=?) in the abstract - rec to add this into results below.  Also, clarify in the abstract, line 22 that environmental were compared to "patient" cultures or some other term rather than "clinical".  

Page 1, line 44 - elaborate on why those specific 4 were chosen as focused bacteria. Virulence +/- incidence in your institution?

In your study design or microbial sampling section - clarify whether your 2 ICUs used were mixed surgical/medical or a specific patient type (e.g. surgical, neuro, cardiac).

Page 2, line 72 - clarify what your standard time frame for room cleaning is. Example: every 24 hrs, after patients are discharged out of the room, some other frequency? Also consider including if/how your room cleaning changed with regard to COVID-19. Example: were the rooms cleaned more or less often based on isolated patients, staff shortages, etc..?  This will help to assess your results on page 4 and why there was a wide variation of 15 to 45 hours in between cleaning and environmental sampling.

Consider including all the positive cultures and sources for all patients (not just those with crossovers) in the results after your total environmental organism listing.  Also, page 5, line 170 - consider listing what organisms were found on the glove sample cultures?

Page 6, line 195, Figure 3 - consider adding lines to each of your listed cross-contaminations for easier reference.  1, 2, 3 etc... and then 7 and 8 on line 7 to represent the 2 patients for a total of 14.

Page 6, line 200 & 201 - consider clarifying what is meant by "positive delay" and "negative delay"

Page 8, line 239 and 240 - consider clarifying the words "stocked" as I am not sure what you are trying to say here.

Some minor considerations :):

Consider changing references in the title and throughout the paper from "real-life" to "real-world"?

Page 2, line 47 - include "colonization" as well as infection in critically ill patients since it was not established that all cultures were associated with infection.

Page 2, line 76 - consider rephrasing for clarity, something like "...; patient cultures were obtained based on routine care and compared against environmental cultures from the same room."

Page 3, line 106 and 112 - consider changing "antibiogram" to "susceptibility" and "susceptibility patterns".  In the US, antibiogram is the term for a broad combined susceptibility for an organism for an entire hospital/unit.  But perhaps those are interchangeable words in France.  If so, ignore.

Page 5, line 169 - change "germs" to "organisms".

Page 4 line 133 and page 6, line 204 - consider changing "sensitive" to "susceptible"

Page 8, line 228 - this may be the appropriate EMBL term "Enterobacters", just double checking that it shouldn't be "Enterobacteriaceae" for the broader micro class.

Reviewer 2 Report

The authors prepared a research manuscript aiming to monitor the microbial community in the hospital environment of the ICUs of a French University Hospital and to explore the potential link between environmental contamination and patients’ colonization and/or infection. Overall, this manuscript signifies an effort to provide insights on such a topic.

I found that the topic of current MS is adequate and acceptable for the journal’s scope. Even more, the manuscript addresses an important topic in the field of infectious disease. The manuscript is well-prepared and written at a proper academic level to meet the high standards of the International Journal of Environmental Research and Public Health. To this end, I recommend the manuscript be considered for publication in the journal. Nevertheless, I have some major and minor concerns/comments related to the contents written in the manuscript.

1.   While the authors have argued for the importance of this topic, the novelty of this manuscript remains overlooked. The introduction section written in the manuscript is really short and quite vague to provide an insight into why the authors decided to carry out this research and to a greater extent, into why this research is important. For example, the authors mentioned in the introduction (lines 44-46) that they decided to focus their attention on four bacteria: S. aureus, E. faecium, A. baumanii, and P. aeruginosa, without proper explanation on why and how they determine their focus on those bacteria. Meanwhile, the results obtained in their study reflected the abundance of other bacteria such as E. faecalis, P. agglomerans, etc.

2.     The authors have carried out a well-designed screening on the samples obtained from the patient or from hospital environments. Indeed, the story provided in the results section is adequate to support the authors' conclusion. However, it remains unclear why particular bacteria, such as E. faecalis, P. agglomerans, present in abundance levels at the locations of the study (as illustrated in Fig. 2). What are the medical implications of these findings? I encourage the authors to discuss this in the discussion section to attract readers' attention and to provoke further studies.

3.  The captions of all figures need to be described after the figures themselves. The current manuscript does not adhere to such a rule thus modifications are strongly required.

4.     Minor: Y-axis of Figure 2 needs to be revised. Do not just put “%”.

5.   How many times the collection and examination of the bacteria were performed? Does the percentage of bacteria illustrated in Fig. 2 have a standard deviation?

6.    Figure 2 is quite small to illustrate the results. The authors need to modify the figure to accommodate better illustration thus readers can easily understand the impact of the results.

7.  Figure 3 is quite confusing. Based on the figure itself, it is quite challenging to understand which of the examination demonstrated the potential contaminations of the environment (E)-patient (P) (positive delay, n=4) and vice versa (negative delay, n=10). One case of P-E-P is shown (September 2021) and do the authors designated this as E-P or P-E or both?

8.     There is no extensive discussion on how the bacteria can present in the environment/inanimate surfaces of the ICU? This is a critical part of the manuscript that needs to be properly addressed.

Reviewer 3 Report

General comments

·      Thank you for asking me to review the manuscript entitled ‘Bacterial cross-transmission between inanimate surfaces and patients in intensive care units under real-life conditions: a prospective point prevalence survey’. Antimicrobial resistance (AMR) is a global health problem that is growing at an unprecedented rate. The environmental contribution to AMR is crucial to the continuing spread of resistant pathogens. Thus, the relevance of this topic.

·      Generally, however, the paper is poorly written and all its sections need to be revised to improve on its quality.

Specific comments

Abstract

·      Line 18: The phase ‘microbial samples’ should be changed to ‘environmental samples’.

·      Line 21: Referring bacteria collected from surfaces that are frequently touched as ‘clinical bacteria’ is innovative but not practical. All the isolates are obtained from environmental samples.

·      Line 25: ‘Bacillus cereus and spp (6.3%)’ can be modified. Does this mean Bacillus cereus and other spp. Bacillus?

Introduction

·      Line 34: The introduction section is too short and lack information to justify the rationale of this study. The authors should synthesize the existing literature and clearly point out the gaps that their paper is addressing.

·      Line 35: The authors should use more standardized terminology. For example, healthcare-associated infections can replace ‘healthcare-associated pathogens’.

Methods

·      Line 50: The authors should properly describe the study design. Was this a cross-sectional study? The author should be clear and concise in their description of the study design.

·      Line 52 and 53: …, and interrupted three times, due to waves of Covid-19 52 pandemic in France (in March-June 2020, October-December 2020, and April-June 2021)’ should be moved into the limitation section. I don’t think it is necessary in the methods section.

·      Line 50: Some statements are confusing. For example,’ this study included one room in two ICUs at the Edouard Herriot Hospital, a 1160-bedded university hospital in Lyon, France’. Does this imply that one room is selected in each of the ICU?

·      I suggest the authors provide more details on the method section. The authors should describe the study setting. How was the sample size determined? How were the ICU locations selected?

Results

·      Line 122 to 125:  The statement ‘Samples were collected between 15 and 45 hours after the last 122 room cleaning (22.2 ± 6.0 hours on average). The patient was present in the room at the 123 time of sample in 85.7% of cases. Nobody else was in the room during sampling, except for 6 occasions with the presence of 1 or 2 staff members. The sample was considered positive when at least one bacterium (of interest or not) was detected’ should be moved to the methods section. This is not a result but a description of the sampling technique and should be moved to the methods section.

·      There is a rule in microbiology. After writing the organism’s name in full, you abbreviate the genome (first) name whenever you write it in the same text. For examples, Pseudomonas aeruginosa can be written as P. aeruginosa after its first appearance. The authors should apply this to all the isolates.

·      Lines 133 and 134: The statement ‘They all had a sensitive profile, except for one isolate of Pseudomonas aeruginosa imipenem-resistant and with an intermediate resistance to meropenem’ is confusing. The authors should state the sensitivity to the antibiotics which were tested for the readers to have a clear understanding.  

·      There is no figure or table to indicate the resistance patterns. This information should be added to improve on the quality of the manuscript. The authors can categorize the bacterial isolates using standardized international classification system (i.e, ESBL, CRE, carbapenem-resistant non-lactose fermenting gram-negative bacilli, carbapenem-resistant Acinetobacter baumannii (CRAB), carbapenem-resistant P. aeruginosa (CRPA), MRSA, VRE Etc.,)

·      The fonts for Figure 2 are small and as such the figure is unreadable to me. The authors can assess and do the needful.

·      Line 161 to 170: The statement in line 161 to 170 should be moved to the methods section. In the result section, the authors have to focus on describing the results and not the method applied to ensure quality results.

·      Line 222: ‘Interestingly, only one bacterium isolated in the environment was resistant to antibiotics’…. This statement is vague. To provide more clarity, the authors should define the antibiotic resistance patterns.

·      I’m not sure about the use of the word ‘point prevalence survey’. The authors should provide an explanation on this.

Discussion and conclusion

·      In my opinion the discussion of the result obtained from this study is very shallow and disjointed. The authors have taken a ‘one health’ approach to describe the interactions of bacteria in human-environment interface. This opportunity can be explored by the authors in discussing the results of this paper.

·      In some cases, the authors just restated the results with little or minimal discussion around them.

·      The statement in Line 240 to 242 is a recommendation and can be considered in the conclusion section

·      Line 257 to 263 is a repetition of the result. The authors can consider revising this

·      The conclusion should not contain citations.

·      The authors should define the main results in the conclusion and the implications of the results but this is not the case in this context.

Round 2

Reviewer 3 Report

The manuscript can be accepted in the present form

This manuscript is a resubmission of an earlier submission. The following is a list of the peer review reports and author responses from that submission.